# Application of Metabolomics in Fungal Research

**DOI:** 10.3390/molecules27217365

**Published:** 2022-10-29

**Authors:** Guangyao Li, Tongtong Jian, Xiaojin Liu, Qingtao Lv, Guoying Zhang, Jianya Ling

**Affiliations:** 1School of Pharmacy, Shandong University of Traditional Chinese Medicine, Jinan 250355, China; 2Hospital of Shandong University of Traditional Chinese Medicine, Jinan 250355, China; 3State Key Laboratory of Microbial Technology, Shandong University, Qingdao 266237, China

**Keywords:** metabolomics, fungi, application

## Abstract

Metabolomics is an essential method to study the dynamic changes of metabolic networks and products using modern analytical techniques, as well as reveal the life phenomena and their inherent laws. Currently, more and more attention has been paid to the development of metabolic histochemistry in the fungus field. This paper reviews the application of metabolomics in fungal research from five aspects: identification, response to stress, metabolite discovery, metabolism engineering, and fungal interactions with plants.

## 1. Introduction

Metabolomics is an emerging omics technology following genomics, proteomics, and transcriptomics. The concept of metabolomics first originated from metabolomic profiling proposed in 1971 by Devaux et al. [1]. Metabolomics was put forward as a group of metabolites in organisms by Oliver et al. in 1998 [2]. Nicholson raised the concept of metabolomics on the basis of a statistical analysis of NMR spectroscopic data from mouse urine. These data are defined as “a quantitative measurement of the dynamic multiparametric metabolic response of living systems to pathophysiological stimuli or genetic modifications” [3]. Traditional metabolomics is divided into targeted metabolomics and untargeted metabolomics. Targeted metabolomics is the measurement of a defined set of chemically characterized and biochemically annotated metabolites, usually focusing on one or more relevant metabolic pathways [4]. Recently, it has been subdivided and further developed into widely targeted metabolomics, pseudotargeted metabolomics, quasi-targeted metabolomics, LM precision targeted metabolomics, etc. Although the above methods all use the MRM mode for mass spectrometric data acquisition, widely targeted metabolomics and pseudotargeted metabolomics are performed by qualitatively and relatively quantifying the target through substances in the local library (established on the basis of partial standards, untargeted data, literature data, etc.), LM precision targeted metabolomics can absolutely characterize the substances corresponding to all standards. Even in combination with external standard methods, absolute quantification of metabolites in a sample can be achieved. However, untargeted metabolomics analyzes all measurable metabolites in a sample. The aim was to measure metabolites in the samples whenever possible [5]. According to different research objects and purposes, metabonomics can be divided into four levels: metabolic fingerprinting analysis, metabolic target analysis, metabolic profiling, and metabolomics [6].

Metabolomics, through modern instrumental analytical methods with high throughput, sensitivity, and resolution, combined with chemometric methods, analyzes the change law of metabolites after stimulation or interference in biological systems. Its focus is more on small-molecule metabolites with relative molecular weights of less than 1000 in biological tissues or cells and is often used to study plant and microbial systems [7,8,9]. The existence time of fungi on the Earth is unknown, and no definite conclusions can be drawn about their origin. Fungal cells do not contain chloroplasts and plastids; they are typical heterotrophs with parasitic or saprophytic patterns. The latest research speculates that there are as many as six million species of fungi worldwide, of which more than 600 are closely related to humans. They can participate in the formation of the human micro-ecosystem as resident fungi or cause diseases as pathogens. Fungi profoundly affect human health, agriculture, biodiversity, natural ecology, industry, biomedicine, etc. [10,11,12]. The application of metabolomics to different research fields of fungi, such as the classification and identification of fungi, metabolic pathways of fungi, the discovery of fungal natural products, and plant–fungal interactions [13,14,15,16,17,18], can help researchers entirely mine the potential of fungi. This article mainly reviews the research methods of metabolomics and the latest progress of metabolomics in various research fields of fungi (Table 1), aiming to promote further research on fungal metabolomics.

## 2. Fungal Metabolomic Approaches

The technical route for metabolomic research in fungi mainly involves three main processes: sample preparation, data collection and processing, and analysis (Figure 1). Sample preparation can affect not only the observed metabolite content but also the biological interpretation of the data. Therefore, appropriate sample collection and preparation steps are required to avoid interfering with the efficient metabolomics analysis. Currently, studies on sample preparation in biological fluids, tissues, mammalian cells, and plants have been relatively comprehensive, but there are few reviews related to fungal sample preparation strategies [61,62,63,64]. An ideal sample preparation method for metabolomics should be as simple, rapid, highly selective, and reproducible as possible and capable of quenching to determine the actual metabolic composition at the sampling time. General sample preparation steps include rapid sampling, quenching, and sample extraction [65].

### 2.1. Rapid Sampling

Rapid sampling from fermentation tanks or shake flasks is the first step in performing fungal sample preparation, since 1969 when Harrison et al. first attempted rapid sampling from small-scale laboratory bioreactors [66], to Iversen and Theobald et al., who developed iterations of sampling systems with minimal dead volume for manual feeding [67,68]. Fast sampling systems are currently characterized by motorized sampling, high frequency, negligible dead areas, and efficient inactivation [69,70,71]. The technology developed by van Gulik used adenine nucleotide as an indicator to analyze its dynamic response to changing glucose concentration and quickly sample yeast metabolites. It has a very high sampling frequency, which can also ensure long-term sterility [72]. Although the sampling method developed by Hannes Link et al., which directly injects fungi such as yeast into high-resolution mass spectrometers for real-time metabolomic analysis, achieves automated detection of target compounds [73], this approach is expensive and impractical for the majority of researchers.

### 2.2. Quenching

The purpose of quenching is to rapidly stop various metabolic activities within the cells, inactivate enzymes, keep the metabolite content stable, and reduce the degradation of metabolites. The cold methanol quenching method is a standard method used in the past, and the research on cell quenching mainly focused on different methanol content and quenching temperature. Specifically, 60% (*v*/*w*) methanol quenching at −40 °C is one of the standard methods for microbial metabolomics research. However, this method causes the leakage of cellular metabolites [74,75]. So far, cold methanol use for quenching has remained a controversial topic. Due to the advantage of a high extraction efficiency of representative intracellular metabolites by cold methanol quenching, some researchers have continuously improved the original method of cold methanol quenching proposed by de Koning et al. According to the degree of metabolite leakage depending on the exposure time, temperature, and nature of the methanol solution, under fast sampling equipment, setting the sample/quencher liquid (*v*/*v*) ratio to 1:5 or less for quenching in pure methanol at ≤−40 °C effectively prevents metabolite leakage [76]. Of course, other quenching methods are also under continuous development. Rapid filtration of liquid nitrogen is more suitable for cell quenching than cold methanol quenching, with minimal damage to cell integrity and improved recovery of intracellular metabolites [77]. However, ice crystals produced by the freezing of liquid nitrogen can potentially damage the cell membrane, which also leads to the leakage of metabolites inside the cell. Quenching using strong acids such as perchloric acid can lead to the degradation of some compounds in a strong acid environment with severe metabolite reduction [78]. It seems that the perfect quenching method is challenging to achieve. Although Meinert et al. believe that the quenching method of methanol quenching solution (60%, −40 °C) has no metabolite leakage, we can still find that the metabolite investigated using this method is insufficient and far from reaching the level of no leakage for all detection indicators [79]. Moreover, most of the quenching studies focused on Gram-positive bacteria [80], Gram-negative bacteria [81], yeast cells [82,83], etc., whereas research on filamentous fungi is rare.

### 2.3. Sample Extraction

How to extract the quenching sample is a critical step in the sample preparation stage. Furthermore, the ideal extraction should not change the metabolites’ physical properties and chemical structure and should maximize the sequestration of the metabolite content. Current extraction methods can presumably be grouped into physical and chemical categories. Physical methods often use homogenizers, ultrasonic, microwave, and other tools. Chemical methods mostly use acid, base, water, methanol, ethanol, and different ratios of water and organic solvents, mixed solvents, etc. Researchers can choose different solvents depending on different samples. As mentioned above, strong acids and bases can cause severe metabolite leakage resulting in a low recovery rate of final metabolites. Although this is a classical extraction method, using strong acids and bases such as perchloric acid is still not recommended. Solvents with the highest extraction efficiency can be selected experimentally. Three sample preparation methods and five solvent mixtures of *Mortierella alpina* were evaluated using gas chromatography/mass spectrometry (GC–MS) [84]. The results showed better reproducibility and recovery of lyophilized. Methanol/water (1:1) was more effective in extracting metabolites of *Mortierella alpina*. Compared with biphasic extraction at different pH with methanol extraction, which is easy and fast and suitable for the extraction of metabolites from *Phanerochaete chrysosporium*, biphasic extraction at different pH is more suitable for target analysis [85]. Of note, despite the high efficiency and recovery of metabolites extracted by supercritical fluids, partially unstable metabolites may undergo decomposition due to the pressure ranging between 200 and 500 bar [86].

### 2.4. Instrumental Analysis Methods

Commonly used metabolomics analysis methods require collecting raw data after sample quenching and extraction. Currently, several analytical methods exist for qualitative and quantitative analysis of metabonomic extracts in metabonomic research. The commonly used analytical methods for fungal metabolomics include gas chromatography/mass spectrometry (GC–MS), liquid chromatography/mass spectrometry (LC–MS), nuclear magnetic resonance spectroscopy/mass spectrometry (NMR–MS), capillary electrophoresis/mass spectrometry (CE–MS), and matrix-assisted laser desorption ionization mass spectrometry (MALDI-MS). GC–MS is often used to analyze substances with excellent thermal stability and volatility, allowing simultaneous analysis of sugars, amino acids, phosphorylated metabolites, organic acids, lipids, amines, and other compounds. It exhibits extraordinary robustness, excellent separation ability, high selectivity, effective sensitivity, and reproducibility [87]. However, there are also some disadvantages, as nonvolatile compounds require derivatization. The main derivatization methods are silanization, acylation, alkylation, and esterification. Koek et al. established a GC–MS spectrometry metabonomic analysis technology suitable for microorganisms and verified the method with various microorganisms [88]. The results showed that the method has good repeatability, effective reproducibility, and fast linear regression characteristics. It can be used for the metabonomic analysis of various components of microorganisms, such as alcohols, aldehydes, amino acids, fatty acids, organic acids, sugars, purines, pyrimidines, and aromatic compounds. HPLC reduces the complexity of samples and offers several advantages, such as simple preparation, high sensitivity, signal reproducibility, minimal noise, and high qualitative and quantitative ability. It is helpful for thermally labile compounds, nonvolatile compounds, polar compounds, and compounds that are macromolecules. With the development of high-performance liquid chromatography (HPLC) and ultra-performance liquid chromatography (UHPLC), the resolution of peaks was improved, and the speed of analysis was accelerated [89]. NMR techniques have the advantages of high reproducibility, accurate quantification, simple sample preparation, measurable analytes in various solvents, clear identification of unknown metabolites, and complete metabolite detection. The disadvantage is low sensitivity, which severely limits the use of NMR in metabolomics [90]. Capillary electrophoresis techniques are relatively new and less applied analytical methods, mainly for studying molecules, but they are preferred when dealing with highly polar, charged metabolites [91]. They allow rapid and high-resolution analysis of charged metabolites such as nucleic acids, amino acids, carboxylic acids, and sugar phosphates. Each analytical tool has advantages and disadvantages. A single analytical platform tool cannot directly and precisely characterize or quantify thousands of small-molecule metabolites involved in fungal metabolic processes. The right combination of tools is often needed depending on the experimental situation to better analyze the target fungi.

### 2.5. Data Processing and Analysis Methods

The original data obtained by the analytical instrument cannot provide a clean and comparable metabolite spectrum. Therefore, the original data must be preprocessed and generally completed in the experimental system. This mainly includes noise reduction and baseline correction, peak detection and deconvolution, normalization, and data standardization [92,93]. The classical analysis method is to use a single variable, i.e., parameter by parameter, or to use multivariable techniques to evaluate group differences. Although the univariate analysis method is simple and convenient, it cannot accurately distinguish the groups when the difference is small. Multivariate analysis can be used to analyze the changes in single metabolites between different groups and the dependent structure of individual molecules [94]. Multivariate analysis can be divided into two categories; one is the unsupervised learning method, which classifies the original data directly, including principal component analysis (PCA), hierarchical clustering analysis (HCA), and self-organizing maps (SOMs). The other is the supervised learning method, i.e., learning the training samples with a given sample label, such as partial least squares discriminant analysis (PLS-DA), partial least squares discriminant analysis based on orthogonal signal correction (OPLS-DA), artificial neural network (ANN), and support vector machine (SVM). Among them, PCA, PLS-DA, and OPLS-DA are the most frequently used multivariate statistical analysis methods in the field of metabolomics [95,96,97,98]. The generally used analysis software includes MetAlign [99], MZmine [100], XCMS [101], Metabolomic Analysis and Visualization Engine (MAVEN), Metabolite Biological Role (MBRole), MetaCoreTM, MetaboAnalyst, InCroMA, and 3Omics [102,103,104].

## 3. Application of Metabolomics in the Field of Fungal Research

### 3.1. Application of Metabolomics in Classification and Identification of Fungal Research

Morphological methods, as the traditional fungal classification method [105], have some limitations, i.e., the classification from appearance characteristics is affected by the fungal growth environment, similar morphology, and other factors, thus affecting the accuracy of classification. Genomic DNA/DNA hybridization [106,107], ribosomal typing [108,109], multilocus sequence typing [110,111], ITS rDNA sequences [112], and lipid profile analysis [113] are popular methods in fungal taxonomic identification. Chemical taxonomy was initially considered complementary to morphological methods based on primary and, more often, secondary metabolites. However, with advances in HPLC and mass spectrometry, the application of metabolomic chemotaxonomic in fungal taxonomic identification has progressed considerably (Table 2). Kang et al. analyzed the secondary metabolites of seven species of *Trichoderma* (33 strains) using its sequence and metabolome-based chemotaxonomic comparison. They found that the chemical taxonomy based on secondary metabolites was more accurate than its sequence and identified an unknown group of *Trichoderma* [114]. Chen used HPLC fingerprinting combined with stoichiometric analysis of *Ganoderma*
*lucidum* fruiting bodies and screened four marker components as discriminative variables to distinguish *Ganoderma lucidum* [115]. However, Wen et al. believed that the identification of distinguishing *Ganoderma lucidum* with the NMR metabolomics method was less time-consuming and faster, which was in line with the quality control of large-scale production. Through NMR metabolomics, labeling choline and propionic amino acids as discriminating variables not only successfully differentiated between Chinese and Korean *Ganoderma lucidum* but also differentiated the cultivated origin of Chinese *Ganoderma lucidum* [116]. This method of taxonomic identification based on the specific metabolites of fungal species effectively avoids the limitation of low accuracy of traditional methods and identifying fungi from different regions or even different growth stages.

Nevertheless, metabolome-based chemotaxonomy also has drawbacks. First, how to overcome the problem of finding specific markers from a plethora of metabolites for optimal biological interpretation is still unanswered. Second, changes in various environmentally relevant regulatory genes may not affect the expression of taxonomic-related genes. Lastly, while it is desirable to analyze metabolites of specific organelles, how organelles can be isolated while maintaining a structural, metabolic state remains unattainable.

### 3.2. Application of Metabolomics in the Study of Fungal Response to Stress

Fungi can survive only under various stress reactions such as ionizing radiation, hydration activity, acid–base environment, hypoxic stress, solar ultraviolet radiation, agricultural and industrial pollutants, biological stress, nutrient stress, oxidative stress, heat stress, and cold stress to survive. Ecological metabolomics studies changes in endogenous metabolites produced by biological systems that are affected by environmental factors [123]. Using metabolomics to study fungi, we can understand how fungi respond to stress when they grow or infect their hosts. It is helpful to optimize the application of fungi in biotechnology, improve the environment, and even prevent fungal diseases.

Environmental stress, i.e., abiotic stress, is usually the most severe situation faced by fungi. Oliveira et al. found that the nutrient content of low-molecular-weight metabolites of wild mushrooms is higher than that of indoor cultivated ones [19], perhaps because the functions of the low-molecular-weight metabolite gene clusters of these mushrooms and their relative expression have differences in the two environments. Metabolomic analysis of the white rot basidomycete *Phanerochaete chrysosporium* under air and 100% oxygen revealed that three metabolites associated with the oxygen stress response were veratryl alcohol (VA), threonate, and erythronate. High concentrations of ROS can directly activate the VA synthesis pathway, and the intracellular oxygen concentration is significantly elevated. In contrast, threonate and erythronate resistance to hyperoxia is a process of progressive gradient accumulation [20]. When the fungal *Alternaria* sp. MG1 grown on grape interiors was subjected to starvation treatment, the shikimate pathway and the phenylpropanoid (PPPN) pathway were strongly activated, and relevant metabolites such as resveratrol were significantly upregulated [21]. The metabolic profiles of *Cryptococcus neoformans* changed under Cu stress, and the differential metabolites were mainly related to the metabolism of amino acids and carbohydrates. Replacing the carbon source with glycerol and ethanol can counteract the toxic effect of copper on *Cryptococcus neoformans* and improve urea clearance [22]. Yan et al. analyzed the metabolites of *Pleurotus ostreatus* under different heat stress times (6, 12, 24, and 48 h) for dynamic metabolite changes. They found that the contents of metabolites such as amino acids, nucleotides, and lipids showed an increasing trend with increased heat stress time [23]. Zhao et al. found that *Volvariella volvacea* showed little resistance to low temperatures. Nevertheless, under chilling stress, the relative levels of compounds such as amino acids and organic acids inside *Volvariella volvacea* increase significantly, and soluble sugars such as sorbitol are induced to be produced, improving its osmoregulatory capacity [24]. Interestingly, *Aspergillus aculeatus*, under drought and heat stress, increased the accumulation of amino acids and sugars and enhanced the total photosynthesis of tall fescue, resulting in a vastly improved ability of tall fescue inoculated with *Aspergillus aculeatus* to resist cold and heat stress [25].

*Aspergillus flavus* showed significant changes in carbohydrates, sulfur-containing amino acids and their derivatives, fatty acids, etc. under drought stress [124]. 1-Nonanol can destroy the integrity of the cell membrane in *Aspergillus flavus* and affect mitochondrial function, which induces apoptosis in *Aspergillus flavus* [26]. *Aspergillus niger* resisted copper stress by converting sorbitol from glucose to produce a large amount of mannitol [27]. When *Aspergillus niger* was exposed to 5% ethanol stress, its growth amount was about 70% less than that under normal growth conditions [28]. Using untargeted metabolomics to study its reaction mechanism, it was found that TAG, DAG, and hTAGs significantly accumulated. These neutral glycerolipids were previously believed to be associated with the fungi’s exposure to abiotic stress factors [125,126]. Whether glycerides in the response of *Aspergillus niger* strain Es4 to ethanol stress can be used as a new response of the fungus to ethanol stress still needs further confirmation. Hammerl et al. further established a differential offline LC–NMR (DOLC–NMR) method to qualitatively and quantitatively analyze metabolic changes in *Penicillium roqueforti* when L-tyrosine levels are perturbed. Twenty-three metabolites were affected by the amino-acid perturbation method, among which the amino-acid degradation products 2-(4-hydroxyphenyl) acetic acid and 2-(3,4-dihydroxyphenyl) acetic acid were significantly upregulated [127]. Jiang et al. found that treatment of *Ganoderma lucidum* with methyl jasmonate (MeJA) for 24 h was the optimal condition to induce the biosynthesis of *Ganoderma lucidum*. MeJA induction can lead to metabolic rearrangements in *Ganoderma lucidum*, inhibit its normal glucose metabolism, energy supply, and protein synthesis, and promote cellular secondary metabolic production [29]. After treatment with tributyltin (TBT), the mycelial morphology of *Cunninghamella echinulata* changed, the metabolic activity was inhibited, and glycolysis and the TCA cycle were dysregulated. This fungus can eliminate the hazard of tributyltin compounds to the organisms by accumulating amino acids with antioxidant functions, such as betaine, proline, and GABA, to recover from the toxic TBT environment [30].

In contrast, benzoic acid derivatives such as sodium benzoate, p-aminobenzoic acid, and p-methylbenzoic acid all promote lipid synthesis in *Penicillium sr21*. Furthermore, 200 mg/L p-aminobenzoic acid even promoted glucose catabolism during glycolysis, increased the mevalonate pathway, weakened the tricarboxylic acid cycle, and promoted the production of tetrahydrofolate and NADPH [31]. The antibacterial polycationic peptide ε-poly-l-lysine (ε-PL) enhances the freeze–thaw tolerance of industrial *yeast* by promoting cell membrane-associated fatty-acid synthesis before freeze–thaw and promoting alglucan biosynthesis and glycerophospholipid metabolism after freeze–thaw [32]. From the above studies, it is easy to see that both biotic and abiotic stresses involve a variety of metabolic pathways in fungi, the most important of which are the metabolism of amino acids and their derivatives, glycolytic pathways, etc. Significant marginal changes in the levels of metabolites such as sugars, nucleotides, and lipids are the main mechanisms of their dynamic regulation.

### 3.3. Application of Metabolomics in the Discovery of Fungal Metabolites

A large number of metabolites such as primary and secondary metabolites exist in fungi. Primary metabolites are monomers synthesized by primary metabolisms, such as monosaccharides or monosaccharide derivatives, nucleotides, vitamins, amino acids, fatty acids, and various macromolecular polymers composed of them, including proteins, nucleic acids, polysaccharides, lipids, and other essential substances. Secondary metabolites refer to substances synthesized by fungi in which primary metabolites serve as precursors with no clear function in their life activities, such as gibberellins, penicillins, aflatoxins, and cordycepin [128]. Fungi can provide diverse and unique secondary metabolites, making them potential drug sources. Traditional assays can easily lead to the rediscovery of known compounds. With the advancement of analytical technology platforms, MS-based metabolomics workflows are mainly suitable for screening hundreds of natural products simultaneously for dereplication studies and extractions of bioactive compounds, which is of great benefit for the comprehensive exploration of potentially useful secondary metabolites. In addition to drug discovery, screening of bioactive compounds or discovery of unknown fungal metabolites that play critical roles in host fungal interactions are also required to identify fungal secondary metabolites. A metabolomics approach can aid in discovering and detecting novel metabolites in fungi (Table 3).

Cordycepin is considered to be an important marker for the identification of *Cordyceps sinensis*. While nucleosides such as thymine, uracil, adenine, and guanosine are also the main substances used by researchers to identify and analyze *Cordyceps sinensis*. Mishra reidentified the nucleobases in samples of *Ganoderma lucidum* and *Cordyceps sinensis* by HPLC–MS and found that both had abundant nucleosides. Furthermore, the water extract of *Ganoderma lucidum* and the ethanolic extract of *Cordyceps sinensis* had the highest nucleobase content [33]. Joshi et al., for the first time, identified the presence of cordycepin using ion mobility mass spectrometry (IMMS), which provided a new method for identifying oridonin [34]. The growth of Cordyceps pupae can be divided into the first to the third stage of growth and the fourth stage of senescence. Principal component analysis found an obvious separation between the first and fourth stages of cordycepin, indicating that cordycepin was significantly enriched in the senescence stage of fruiting bodies [35]. Furthermore, cordycepin, the contents of amino acids and carbohydrates such as glucose, xylitol, and mannose were also obviously increased. The biosynthesis of cordycepin may be regulated by the glutamine and glutamate metabolic pathways. Although *Cordyceps militaris* and *Cordyceps sinensis* belong to the same family as Clavicipitaceae, and *Cordyceps militaris* is even called “*northern Cordyceps sinensis*” in China, they have drastically different nutritional contents, which Chen et al. confirmed from the metabolite level. The results of utilizing LC–MS technology to analyze natural *Cordyceps sinensis* and artificial cultured *Cordyceps militaris* showed significant metabolomic differences between them [36]. Similarly, the chemical composition of *Cordyceps sinensis* and *Cordyceps militaris* cultured with *tussah pupae* was compared, and 25 differential metabolites were found, involving 16 metabolic pathways such as histidine metabolism. *Cordyceps sinensis* has many healthy nutrients, especially amino acids, unsaturated fatty acids, peptides, and mannitol. Moreover, the superior hemostatic activity and the antioxidant capacity of *Cordyceps sinensis* cultured with *tussah pupae* suggest its extreme clinical value as an affordable alternative to oridonin [37]. *Cordyceps militaris* strains were inoculated on germinated soybean (GSC), and the yield and biological activity of GSCs reached the highest after 1 week. Compounds 1–4, which were highly abundant in GSCs, were identified as four novel isoflavone methyl glycosides (daidzein 7-o-β-D-glucoside 400-o-methide, glycitein 7-o-β-D-glucoside 400-o-methide, genistein 7-o-β-D-glucoside 400-o-methide, and genistein 40-o-β-D-glucoside 400-o-methide) [38]. Apparently, mixed coculture is a good way to improve the nutrients of *Cordyceps militaris*. Coculture of fungi is often beneficial to induce purposeful fungal differentiation, affect the content of metabolites, and produce multiple metabolic pathways. *Coculture of Coriolus* versicolor and Ganoderma lucidum with 62 newly synthesized or high-yielding features compared to monoculture. Two new xylosides (compounds 2 and 3) were included. Compound 2 was further identified as N-(4-methoxyphenyl) carboxamide 2-O-β-D-xyloside, which increases the viability of the BEAS-2B human immortalized bronchial epithelial cell line. 3-Phenyllactic acid and orsellinic acid were first detected in *malate bacilli* [39]. However, fungal interactions also produce antagonistic inhibition. The coculture of *Aspergillus oryzae* and *Zygosaccharomyces rouxii* reduced the amounts of imidazoleacetic acid, phenylpyruvic acid, and n-formyl-l-aspartic acid, taurine, and glycolic acid [40]. Obviously, coculture inhibited the growth of *Zygosaccharomyces rouxii*.

*Agaricus bisporus* is a worldwide edible mushroom. The surface browning of mushrooms is one of the major factors affecting consumers’ purchase. The nutritional value of *Agaricus bisporus* changed after UV irradiation, with 47 compounds increasing in concentration and 72 compounds decreasing in concentration [131]. Looking at the difference between browning tolerant cultivars and common *Agaricus bisporus* cultivars at the metabolic level, genes such as AbPPo were found to be involved in the regulation of mushroom browning, and higher levels of organic acids, such as butyric acid, were found in brown tolerant *Agaricus bisporus* cultivars [41]. In addition, the pH value and alginate content concentration also affected the activity of AbPPo. Lower pH levels inhibited the expression of AbPPo, and high alginate concentrations may be beneficial for maintaining the activity of AbPPo. The browning of filamentous mushrooms appears to be different. Yu et al. suggested that phenylpropanoid biosynthesis and tyrosine metabolism may promote the browning of filamentous fungi. In addition, dopa melanin accumulation may also be one of the causes for the browning of *Flammulina velutipes* [42].

Endophytic fungi are ubiquitous in the plants body and should be actively exploited to utilize these resources, whether harmful to the plant or not. Hawary et al. isolated a butenolide derivative from the endophytic soybean fungus *Thra terreus*, Aspergillide B1, as well as 3a-hydroxy-3,5-dihydromonacolin L [43]. Using computer-aided technology such as CADD, they suggested that Aspergillide B1 and 3a-hydroxy-3,5-dihydromonacolin L are promising candidates for the treatment of COVID-19. Nevertheless, the speculation is limited to computer-assisted approaches, it lacks pharmacological experimental validation, and whether it is effective remains to be proven. Tawfike et al. isolated the endophytic fungus *Aspergillus flocculus* from *Markhamia Platycalyx*, and the secondary metabolites cis-4-hydroxymellein, 5-hydroxymellein, diorcinol, bo-tryoisocoumarin A, and mellein had anticancer activity and inhibited the growth of the chronic leukemia cell line K562 3-hydroxymellein. Moreover, diorcinol can suppress sleeping sickness caused by the parasite Trypanosoma brucei [129]. Kamal et al. successfully predicted two compounds, clodospirone B and demethyl laciodilodine, with good anti-trypanosome effects from the endophytic fungus *Lasiodiplodia theobromae* [132].

The metabolites of fungi would change under different fermentation times. When the fermentation time is too short, the content of the target metabolites might not yet have peaked, whereas, when the fermentation time is too long, the target metabolites might have undergone decomposition. For the first time, Bu et al. showed that anthocyanins could be produced from fungi as a metabolite often thought to exist only in natural plants. They performed a comparative analysis of the metabolome of *Aspergillus sydowii* H-1 on the second and eighth days of fermentation and found significant differences in the production of five anthocyanins, the chalcone synthase gene, and cinnamic acid-4-hydroxylase gene, which may be associated with the synthesis of anthocyanins [130].

Fan et al. applied HRMS/MS feature-based molecular networking technology (FBMN) to determine *Pyr enochaetopsis* sp. They identified proteins A, B, and C in FVE-001 and protein D in FVE-087, four novel decaprenylspirotetraenoic acid derivatives with anti-melanoma activity [133]. FBMN has several functions in the identification and directional separation of stereoisomers. They combined UPLC–Q-TOF-MS with FBMN to discover three novel similar desferriferriferrichrome compounds [44] from wild *Morchella* sp. Le et al. used this approach to investigate the metabolomics of a strain of *Penicillium* mms417 isolated from blue mussel *Mytilus edulis* and obtained five new derivatives of natural fungus pyran-2-one derivatives: 5,6-dihydro-6S-hydroxymethyl-4-methoxy-2H-pyran-2-one, (6S, 1′R, 2′S)-LL-P880β, 5,6-dihydro-4-methoxy-6S-(1′S, 2′S-dihydroxy pent-3′(E)-enyl)-2H-pyran-2-one, 4-methoxy-6-(1′R, 2′S-dihydroxy pent-3′(E)-enyl)-2H-pyran-2-one, and 4-methoxy-2H-pyran-2-one [45]. Combining metabolomics and FBMN, compound features can be highlighted and clustered together to achieve efficient dereplication of compounds, greatly reducing the difficulty of discovering new metabolites.

### 3.4. Application of Metabolomics in Fungal Metabolic Engineering

Bailey defined metabolic engineering as “improving cellular activity by manipulating the enzymatic, transport, and regulatory functions of cells through the use of recombinant DNA technology” [134]. Fungi exhibited a variety of capabilities in industrial applications, including organic acid fermentation, protein production, and secondary metabolism. Advances in genome engineering have expanded the range of potential applications for fungal bioproduction. The development of genetic engineering tools is essential for efficiently utilizing genomic data. Currently, sequencing analyses of many filamentous fungi have revealed an underestimated potential, i.e., the presence of a large number of silent secondary metabolite genes. Metabolomics methods can be used to analyze changes in various metabolites of fungi after DNA recombination.

Huang et al. measured six major metabolites in the isoprene biosynthetic pathway using GC–SIM-MS and detected the changes after gene modification [46]. The roles of the erg9 and CoQ1 genes could be used as targets to aid in redirecting sterol precursors to the phosphorylated isoprenoid pathway. This approach could enhance the understanding of this pathway in many biological systems. To investigate the effect of histone deacetylase activity (HDACi) on the model fungus *Aspergillus nidulans*, Albright et al. analyzed the changes in more than 1000 small molecules secreted by *Aspergillus nidulans*. They found that almost the same number of compounds were upregulated and downregulated more than 100-fold after genetic or chemical reduction of HDACi [47]. Fellutamides, the natural product of *Aspergillus nidulans*, were first detected as a proteasome inhibitor that can be expressed about 100-fold or more upon HDACi induction. When using ionic liquids to stimulate *Aspergillus nidulans*, choline upregulated the primary metabolism of *Aspergillus nidulans*, while 1-ethyl-3-methylimidazolium chloride downregulated the primary metabolism, both of which stimulated the production of acetyl CoA and nonproteinogenic amino acids. Twenty-one of 66 known skeleton genes were upregulated [48].

Interactions between fungi and bacteria cause metabolic modifications in fungi. After undergoing in vitro confrontation culture, the metabolome changes of *Fusarium verticillioides* were much greater than those of Streptomyces sp. Compared with monoculture, many metabolites of *Fusarium verticillioides* were overproduced under resistant conditions compared to *Streptomyces* sp., especially 16 proteinogenic amino acids, inosine, and uridine, which means that the corresponding rate of protein synthesis would be slowed down, resulting in slower growth and less toxigenesis of *Fusarium verticillioides* [49]. Both the environment and the pH affect the biosynthesis of mycotoxins from *Fusarium verticillioides*. After targeted disruption of *Fusarium verticillioides* by the pH-responsive transcription factor PAC1, pH and PAC1 interference were found to affect the biosynthesis of arabitol, mannitol, and trehalose. Trehalose biosynthesis is reduced in PAC1-impaired plants. All three genes are downregulated when PAC1 is perturbed [50]. Using a *Fusarium graminearum* strain deficient for the H3K27 methyltransferase kmt6 to assign metabolites to genes, Ampressa et al. isolated large amounts of fusaristatin A, gibepyrone A, and fusarpyrones A and B from kmt6 mutants by activating silent metabolic pathways through mutations in repressive chromatin modifying enzymes. Triterpenones and trioctanoic acid were found in kmt6fus1 double mutants [51]. GC–EI-MS-based metabolomics has proven to be effective in unraveling the effects of genetic engineering and fungicide toxicity on fungal metabolism. Liu et al. studied the metabolism of *Fusarium graminearum* strains producing low toxins using a metabolome approach based on NMR and GC–MS and found new possible bactericidal targets [52]. The phenotypic observation and significance of nucleobase transporters in *Aspergillus nidulans* tolerance to Boscalid were validated by kalampokis et al. through metabolomic analysis of various biosynthetic pathways and metabolites [53].

Relative to other microorganisms, fungi are more classified. Furthermore, filamentous fungi are quite different from fungi such as yeasts in terms of growth mode and genetic characteristics. Multinucleated filamentous fungi are prone to heterokaryotic transformant phenomena. Thus, gene editing on filamentous fungi requires rapid and efficient manipulation techniques. Metabolomics and the construction of metabolic networks benefit the optimization and improvement of fungal metabolic pathways. Compared with transcriptomics and proteomics, metabolomics is able to more keenly analyze the effects of environmental perturbations or stresses on cells. Because there are cases where environmental alterations do not affect changes at the cellular transcriptional or protein level but can be manifested by metabolites.

### 3.5. Application of Metabolomics in the Field of Plant–Fungal Interaction

As mentioned above, endophytic fungi are ubiquitous in plants in nature. Some endophytic fungi have developed a mutually beneficial symbiosis with the hosts during a long period of evolution. They can regulate the hormone levels of plants, produce secondary metabolites similar to their hosts, assist the host plants in resisting environmental stresses, etc. Others are harmful fungi, and plant diseases caused by harmful fungi pose a significant threat to global food security. Understanding the interactions between fungi and plants is essential for preventing and controlling plant diseases. In the last decade, metabolomics technologies have been widely applied in various research fields on fungal plant interactions, such as identifying fungi, determining the mechanism of infection, and detecting the interaction between fungi and the host. The applications of metabolomics can help us to understand the pathogenesis and plant defense mechanisms of pathogenic fungi and develop effective prevention and treatment strategies for fungal diseases.

The rate of plant primary and secondary metabolite production is limited by its growth cycle, but endophytic fungi can promote the formation of metabolites from parasitic plants, as seen in *Diaporthe phaseolorum* (DP) and *Trichoderma spirale* (TS) during their symbiosis with *Combretum lanceolatum*. DP promotes the biosynthesis of primary metabolites such as threonine, malate, and N-acetylmannosamine of *Combretum lanceolatum*, which are metabolite precursors that have been shown to be bioprotective [54]. In the case of mutually beneficial symbiotic plants with fungi, plants can provide essential nutrients for fungal survival, and fungi mediate host plant defense responses to stresses such as environmental stress. When *Pisolithus tinctorius* was parasitized on *cork oak* roots, the contents of root exudates such as carbohydrates, organic acids, tannins, long-chain fatty acids, and monoacylglycerols were significantly decreased. In contrast, root defense substances such as γ-aminobutyric acid (GABA), a terpenoid, guarantee that the *cork oak* roots can control the proliferative range of *Pisolithus tinctorius* while symbiosing with *Pisolithus tinctorius* [55]. Phytohormones such as salicylic acid (SA) and jasmonic acid (JA) are endogenous regulators used by higher plants to defend against foreign pathogens [135,136]. Metabolic pathways are significantly different in soybean inoculated with *Fusarium Verticillium* compared to normal soybean. Flavonoid contents are significantly higher in soybean inoculated with molds, and Ja induces the synthesis of biomacromolecules such as glycine to enhance soybean resistance [56]. Moreover, mannitol, threitol and trehalose were significantly enriched in *Armillaria luteobuablina*-treated roots [57]. Exogenous threitol could promote the colonization of *Armillaria luteobuablina* in *E. grandis* roots and trigger hormonal responses in root cells, a phenomenon that was not detected in previous studies.

Fungal diseases are not only able to invade host plants initially, but they can also still invade again after rehabilitation. Mainly through spore germination or infecting the orifice through the mycelium, a few fungi can invade directly through the cuticle of plant tissue. Ren et al. performed LC–MS metabolomic analysis of grains infected with *Tilletia controverta* and normal grains (Figure 2). They found that the expression of 9-HODE, prostaglandin D3, caffeic acid, L-phenylalanine, and tetradecanoic acid was significantly upregulated. In infected samples, prostaglandin D3 was a coordinating factor to promote the increase of other metabolites involved in body defense. L-Phenylalanine promotes the synthesis of lignin monomers, caffeic acid, tetraacetic acid, etc., which are antifungal substances produced by cereals. In addition, the content of grain metabolites such as malate, L-proline, and fumarate decreased, indicating that the level of self-metabolism in *Tilletia controverta*-invaded wheat was inhibited [58]. In contrast, when *Tilletia caries* infested wheat, it did not directly change the metabolites of wheat. However, it prompted wheat to change the key metabolites and reduce the defense resistance function of wheat through pathways such as reducing the immune response to the sweet taste of wheat [137].

Metabolomics is more conducive to developing rapid and effective drugs against fungal diseases than traditional chemical methods. In the past, most antifungal disease drugs were synthetic fungicides, based on the international new trend of environmental protection and green health. Developing natural antifungal components is a more reasonable choice. Chen et al. found that pinocembroside (PICB) isolated from *Ficus hirta* Vahl could significantly inhibit mycelia growth of *Penicillium digitatum*, a pathogenic bacterium of citrus green mold disease [59]. Metabolomics studies have shown that PICB alters the morphology of mycelium and *Penicillium digitatum* cells and promotes membrane peroxidation, which may be associated with the disruption of amino-acid metabolism, lipid metabolism, fatty-acid metabolism, TCA cycle, and purine metabolism. In addition, tomatoes were treated with the secondary metabolites 6-pentyl-2H-pyran-2-one and hartstic acid isolated from *Trichoderma* fungi, and metabolites were studied by HRMAS-NMR. Tomato samples treated with *Trichoderma* fungi secondary metabolites had significantly increased levels of acetylcholine, GABA, and amino acids [60]. These are well-known metabolites advantageous for plant growth, illustrating that developing antifungal disease drugs from a certain endophytic fungus is also a feasible approach.

## 4. Conclusions

As can be seen from the above analysis, metabolomics is a potent and effective tool. Through instrument analysis and data processing, we can gain insight into the changes of small-molecule metabolic components in test samples caused by biotic or abiotic factors. Furthermore, they can be associated with related metabolic pathways, metabolic networks, and metabolically related enzyme gene sets, transcriptomes, and proteomes. Metabolomics is widely used in fungal research and can provide a comprehensive and systematic analytical approach for fungal research. With the continuous development and improvement of sample preparation methods and analytical techniques, fungal metabolomics has made great progress in recent years. However, there are still some urgent issues to be solved in fungal metabolomics. For example, there is no standard method to quench and extract fungal metabolites, the data processing is complicated, and the automatic data processing platform technology is imperfect. The fungal metabolomics database is rare and incomplete, which is a crucial factor constraining the further development of fungal metabolomics.

On the other hand, although more than a few thousand metabolites have been identified, this is still only the tip of the iceberg for fungal metabolites. More importantly, many researchers still have an insufficient understanding of the metabolic pathways for fungal metabolomics and are only limited to primary and secondary metabolite studies in fungi. Compared with the application of metabolomics in disease diagnosis and drug research development, fungal metabolomics is still at an early stage of development. It is believed that metabolomics technology will be continuously improved with the continuous development of science and technology. Greater progress will also be made in fungal metabolomics studies. In conclusion, metabolomics has provided new insights into fungal research from different perspectives, which can be tightly integrated with other studies so that metabolic pathways, regulatory responses, and homeostatic mechanisms can be deeply investigated. This contributes to a better and deeper understanding of fungi’s complex interactions and their responses to environmental and genetic changes.

## Figures and Tables

**Figure 1 molecules-27-07365-f001:**
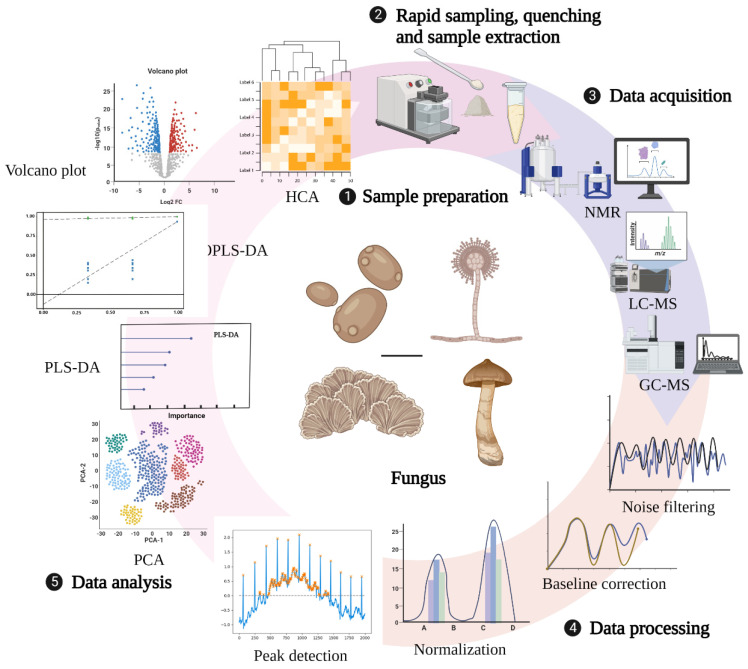
General technical route of metabolomics. Created with BioRender.com (accessed on 28 October 2022).

**Figure 2 molecules-27-07365-f002:**
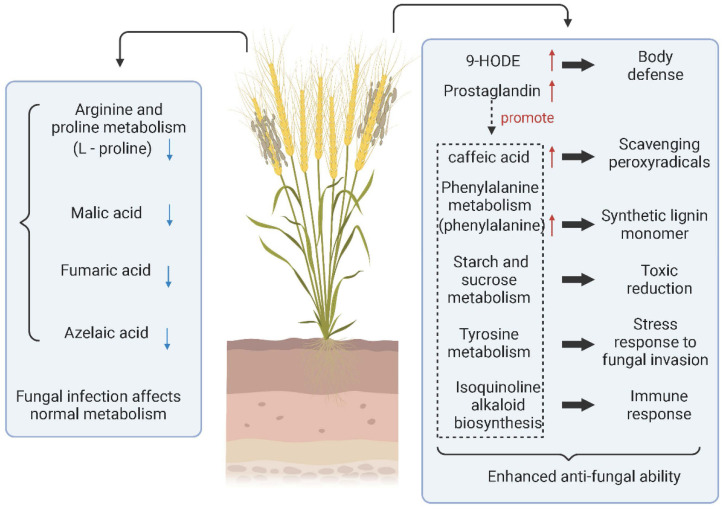
Metabolite changes and metabolic pathways involved in wheat infected with Tilletia controverta. (↑) represents upregulation and (↓) represents downregulation. Created with BioRender.com (accessed on 17 May 2022).

**Table 1 molecules-27-07365-t001:** Application of metabolomics in fungal research in recent years.

Species	Techniques	Nos. of Metabolites	Main Metabolites	Involved Pathway	Ref.
Fungal response to stress					
*Agaricus subrufescens*	UHPLC–MS/MS	38	Ergosterol, agaritine, pyroglutamic acid, vitamin B3, sphingolipids		[19]
*Phanerochaete chrysosporium*	GC–MS	53	Veratryl alcohol, threonate, and erythronate		[20]
*Alternaria* sp. MG1	GC–TOF-MS	239	Amino acid, carbohydrate, xenobiotics, and lipid	PPPN biosynthesis pathway	[21]
*Cryptococcus neoformans*	GC–TOF-MS		Amino acids, carbohydrates	Amino acid and carbohydrate metabolism	[22]
*Pleurotus ostreatus*	LC–Q/TOF-MS	59	Sucrose, dextrin, adenine, uracil, L-glutamine, and L-lysine	glutathione metabolism, sphingolipid metabolism, and some amino-acid metabolism	[23]
*Volvariella volvacea*	LC–Q/TOF-MS	547	Organic acids, fatty acids, amino acids, carbohydrate metabolites, nucleotides	Amino-acid metabolism, carbohydrate metabolism, the TCA cycle, energy metabolism	[24]
*Aspergillus aculeatus*	GC–MS	42	Amino acids, organic acids, sugars, fatty acids, and sugar alcohol		[25]
*Aspergillus flavus*	LC–MS/MS	135		Tricarboxylic acid cycle, amino acid biosynthesis, protein degradation, absorption, mineral absorption	[26]
*Aspergillus niger*	GC–MS		Mannitol and gluconic acid	Mannitol cycle	[27]
*Aspergillus niger*	LC–MS/MS	68	Triacylglycerol, monoacylglycerol, hydroxy-triacylglycerol	Glycerolipid metabolism	[28]
*Ganoderma lucidum*	GC–MS and LC–MS/MS	154/70	L-Malic acid, alpha-hydroxycholesterol, zymosterol, ergosterol	Protein digestion, absorption, purine metabolism, unsaturated fatty acids, fatty-acid biosynthesis, purine metabolism	[29]
*Cunninghamella echinulata*	LC–MS/MS		Protein and amino acid	Purine, amino-acid, TCA, and sugar metabolism	[30]
*Schizochytrium limacinum* SR21	GC–MS	30	Fatty acids, amino acids, organic acids, carbohydrates, alcohols, squalene, cholesterol	Mevalonate, lipid synthesis, and pentose phosphate pathway	[31]
Industrial yeast	GC–MS	59	Trehalose, glycerin acid, fatty acids	TCA cycle, fatty-acid synthesis, glycolysis pathway, arginine metabolism, etc.	[32]
Discovery of fungal natural products					
*Ganoderma lucidum* and *Cordyceps sinensis*	HPTLC–MS	6	Thymine, uracil, adenine, cytosine, guanine and guanosine		[33]
*Ophiocordyceps sinensis*	UHPLC–Q-TOF-IMS	345	Tyrosyl-phenylalanine, 2-phenylethyl beta-D-glucopyranoside and 3′,5′-odimethylmyricetin 3-O-beta-D-2″,3″-diacetylglucopyranoside		[34]
*Cordyceps militaris*	GC–MS	39	Amino acid, nucleosides, organic acids, and sugars	Nucleotide, carbohydrate, and amino-acid metabolism	[35]
*Ophiocordyceps sinensis* and *Cordyceps militaris*	LC–TOF-MS	100	Amino acids, unsaturated fatty acid, peptides, mannitol, adenosine, and succinoadenosine		[36]
*Cordyceps sinensis* and *Cordyceps militaris*	LC–MS	39	L-Tyrosine, 9,10-dihydroxy-12Z-octadecenoic acid and (−)-riboflavin	Histidine metabolism	[37]
*Cordyceps militaris*	LC–ESI-IT-MS/MS and GC–EI-IT-MS		Soyasaponin, pyroglutamic acid, isoflavone methyl-glycosides		[38]
*Trametes versicolor* and *Ganoderma applanatum*		57	N-(4-Methoxyphenyl)formamide 2-O-β-D-xyloside and N-(4-methoxyphenyl)formamide 2-O-β-D-xylobioside		[39]
*Aspergillus oryzae* and *Zygosaccharomyces rouxii*	UHPLC–Q-TOF-MS	32	N-Formyl-l-aspartate, imidazoleacetic acid, taurine, glycocholate, phenylpyruvate	Histidine metabolism, phenylalanine, adenosine kinase, phosphatidylserine synthase homo sapiens, phosphatidylethanolamine scramblase	[40]
*Agaricus bisporus*	UPLC–Q-TOF-MS	40	Organic acids, trehalose	Fatty-acid biosynthesis, tyrosine metabolism, and citrate cycle	[41]
*Flammulina filiformis*	HILIC–ESI(±)-QTOF-MS, LC–MS/MS	107	Melanin, l-dopa (3,4-dihydroxy-l-phenylalanine)	Phenylpropanoid biosynthesis and tyrosine metabolism	[42]
*Aspergillus terreus*	LC–HRMS	18	Quinones, isocoumarins, polyketides		[43]
*Morchella* sp.	UPLC–Q-TOF-MS	50	Fatty acids, peptides		[44]
*Penicillium restrictum* MMS417	UPLC–IT/TOF-MS/MS		Pyran-2-ones		[45]
Fungal metabolic engineering					
*Saccharomyces cerevisiae*	GC–EI-MS		Geranyl diphosphate, farnesyl diphosphate, geranylgeranyl diphosphate, squalene, lanosterol, and ergosterol	Isoprenoid pathway	[46]
*Aspergillus nidulans*	LC–MS		Fellutamide B, antibiotic 1656-G, and antibiotic 3127		[47]
*Aspergillus nidulans*	UHPLC–ESI-HRMS	6	Orcinol, phenoxyacetic acid, orsellinic acid, monodictyphenone, gentisic acid, and caffeic acid	Glycine, serine, and threonine metabolic pathway, glycolysis, and TCA cycle	[48]
*Fusarium verticillioides* and *Streptomyces* sp.	LC–ESI-QqQ	36	Amino acids, saccharides, nucleotides, organic acids, phenol, lipid, and amine	Protein synthesis, Krebs cycle	[49]
*Fusarium verticillioides*	GC–MS	46	Arabitol, mannitol, and trehalose	Fumonisin biosynthesis and trehalose biosynthesis	[50]
*Fusarium graminearum*	LC–MS	22	N-Ethyl anthranilic acid, N-phenethylacetamide, tricinolone and tricinolonoic acid, fusarins, zearalenones, and fusaristatin A		[51]
*Fusarium graminearum*	NMR–GC-FID–MS	45	Sugars, amino acids, organic acids, choline metabolites	Inhibiting glycolysis, tricarboxylic acid cycle	[52]
*Aspergillus nidulans*	GC–EI-MS	86	Carbohydrates, amino acids, and carboxylic and lipid acids, purines and pyrimidines	Amino-acid and carbohydrate metabolism	[53]
Plant–fungal interaction					
Diaporthe phaseolorum, Trichoderma spirale	NMR	20	Threonine, malic acid, and N-acetyl-mannosamine		[54]
*Pisolithus tinctorius*	NMR, FT-ICR	61	Carbohydrates, organic acids, tannins, long-chain fatty acids, monoacylglycerols, gamma-aminobutyric acid (GABA), and terpenoids		[55]
*Fusarium verticillioides*	UPLC–Q-TOF/MS		Isoflavones, jasmonic acid	Phenylpropanoid, flavone metabolic,	[56]
*Armillaria luteobubalina*	GC–MS	117	Sugars, sugar alcohols, amines, or amino acids	D-Threitol synthesis	[57]
*Tilletia controversa*	LC–MS	62	9-HODE, prostaglandin D3, caffeic acid, pyroglutamic acid, tetracosanoic acid		[58]
*Penicillium digitata*	UHPLC–Q-TOF/MS	85	amino acids, lipids, fatty acids, TCA metabolites, galactose metabolites, carbohydrate metabolites, nucleic acids, amino sugars, and nucleotide sugars	Amino-acid, lipid, fatty-acid, and purine metabolism, and TCA cycle	[59]
*Trichoderma* fungi	HRMAS NMR		γ-Aminobutyric acid, acetylcholine, and amino acids		[60]

**Table 2 molecules-27-07365-t002:** Metabolomics for taxonomic identification of fungi.

Fungal Species	Analysis Platform	Extraction Method	Data Processing	Achievement	Ref.
*T. harzianum* *T. aggressivum* *T. virens* *T. longibrachiatum* *T. hamatum* *T. koningii* *T. atroviride*	LC–ESI-MS-MS	The concentrate was pooled into 100 µL of methanol and filtered through a 0.45 umptf filter	Varian MS Workstation 6.9, Vx Capture 2.1, MetAlign, SIMCA-P+ 12.0, Statistica 7	Chemical taxonomy based on secondary metabolite profiling was found to be advantageous over other classification methods	[114]
*Ganoderma lucidum*	NMR spectroscopy	CD3OD and D2O (*v*/*v*, 1:1), 10 mM sodium phosphate, and 0.025% TMSP were mixed and extracted, followed by centrifugation	Matlab, SIMCA-P version 11.0, Chenomx, and Excel	Development of a method to effectively distinguish between national and even regional sources of *G. lucidum* cultivation	[116]
*Rhizoctonia solani*	GC/MS	Derivatization in autosampler vials, upon addition of 80 μL of methoxyamine hydrochloride solution (30 °C, 120 min) and 80 μL of MSTFA (37 °C, 90 min)	ACD/Spec Manager v.12.00, mass spectra matching the National Institute of standards and Technology Library, SIMCA-P 12.0	Characterization and identification of an isolate of *Rhizoctonia solani*	[117]
*Aspergillus*	MALDI-TOF-MS	Bead disruption sample pretreatment followed by centrifugation	BioRad data processing suite	Can be used to unambiguously identify members of the genus *Aspergillus* at the species and strain level	[118]
*Candida species*, *Aspergillus species,* and other yeast genera	MALDI-TOF-MS	Washed yeast cells were fixed by suspension in 50% methanol/water (*v*/*v*) or stored at 4–6 °C for 45 days for subsequent comparative analysis	External alignment was performed using cytosolic picolinic acid A, etc.; MALDI mass spectra were processed using “Data Explorer” (Applied Biosystems), and data were processed in MATLAB	Was used to identify yeast and group strains, as well as follow morphogenesis of *C. albicans*	[119]
*Epichloë festucae*	LC–HR-MS/MS	MTBE, methanol, and water were extracted in two phases, dissolved in 60 µL of methanol/acetonitrile/water (*v*/*v*/*v*, 1:1:12), and centrifuged	The datasets were processed with markerlynx XS for maslynx v.4.1, and the software suite Marvis did the subsequent processing	A genetic approach combined with tandem mass spectrometry was used to identify novel products of secondary metabolite gene clusters and to discover novel Leu/Ile glycoside metabolites	[120]
Wide edible mushrooms	UHPLC–QE Orbitrap/MS/MS	Chloroform and methanol are mixed (2:1 *v*/*v*), then centrifuged	SPSS 16.0 statistical analysis, xcalibur 4.0, ms-dial 4.36, and lipidmaps for identification and quantification of lipids, SIMCA 14.1, metaboanalyst 4.0 follow-up analysis	It is helpful for improving the sensitivity, reproducibility, and accuracy of trace-level analysis of triterpenoids in complex biological samples	[121]
*Ganoderma lucidum* mycelium	UPLC–ESI-HR-QTOF-MRM	Methanol post-extraction filtration	Masslynx 4.1 performed data acquisition, targetlynx quantification, and SPSS 17.0	Highly precise identification and quantification of triterpenoids present in trace amounts in mycelia of *G. lucidum*	[122]

**Table 3 molecules-27-07365-t003:** Novel metabolites discovered and detected using metabolomics.

Structure	Molecular Weight	Molecular Formula	Compound Name	Reference
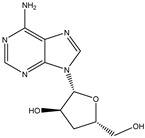	251.2460	C_10_H_13_N_5_O_3_	Cordycepin	[34]
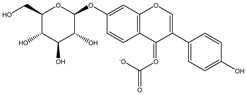	460.3915	C_22_H_22_O_9_	Daidzein 7-O-beta-D-glucoside 4-O-methylate	[38]
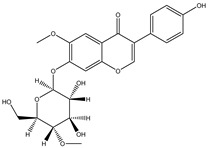	460.4350	C_23_H_24_O_10_	Glycitein 7-O-beta-d-glucoside 4″-O-methylate	[38]
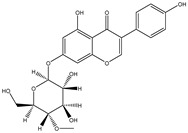	446.4080	C_22_H_22_O_10_	Genistein 7-O-beta-d-glucoside 4″-O-methylate	[38]
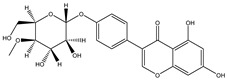	446.4080	C_22_H_22_O_10_	Genistein 4′-O-beta-d-glucoside 4″-O-methylate	[38]
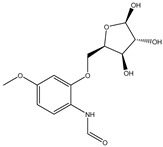	299.2970	C_13_H_17_NO_7_	N-(4-Methoxyphenyl)formamide 2-O-beta-D-xyloside	[39]
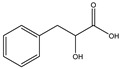	166.1760	C_9_H_10_O_3_	3-Phenyllactic acid	[39]
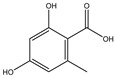	168.1480	C_8_H_8_O_4_	Orsellinic acid	[39]
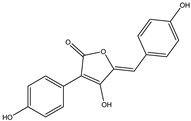	296.2780	C_17_H_12_O_5_	Aspergillide B1	[43]
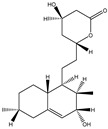	322.4450	C_19_H_30_O_4_	3a-Hydroxy-3, 5-dihydromonacolin L	[43]
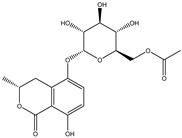	398.3640	C_18_H_22_O_10_	5-Hydroxymellein	[129]
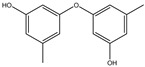	230.2630	C_14_H_14_O_3_	Diorcinol	[129]
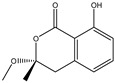	208.2130	C_11_H_12_O_4_	Botryoisocoumarin A	[129]
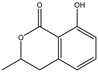	178.1870	C_10_H_10_O_3_	Mellein	[129]
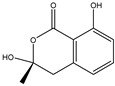	194.1860	C_10_H_10_O_4_	3-Hydroxymellein	[129]
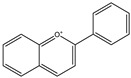	207.2515	C_15_H_11_O^+^	Anthocyanins	[130]
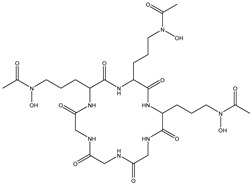	687.7080	C_27_H_45_N_9_O_12_	Desferriferrichrome	[44]
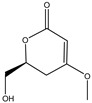	158.1530	C_7_H_10_O_4_	5,6-Dihydro-6s-hydroxymethyl-4-methoxy-2h-pyrene-2-one	[45]
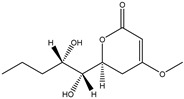	230.2600	C_11_H_18_O_5_	(6S, 1′r, 2′s)—ll-p880 β	[45]
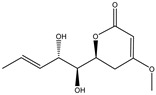	228.2240	C_11_H_16_O_5_	5,6-Dihydro-4-methoxy-6S-(1′S, 2′S-dihydroxy pent-3′ (E)-enyl)-2H-pyran-2-one	[45]
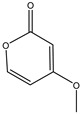	126.1110	C_6_H_6_O_3_	4-Methoxy-6-(1′R, 2′S-dihydroxy pent-3′ (E)-enyl)-2H-pyran-2-one	[45]
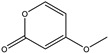	126.1110	C_6_H_6_0_3_	4-Methoxy-2H-pyran-2-one	[45]

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
