# Peer review of "Application of Metabolomics in Fungal Research"

_molecules, 2022, doi:10.3390/molecules27217365_

Round 1
Reviewer 1 Report (Previous Reviewer 1)
The authors significantly modified the manuscript, addressing my previous comments. Thus, it is suitable for publication in this journal.
Author Response
Thank you very much for your support and encouragement.
Reviewer 2 Report (New Reviewer)
The manuscript provides a good overview about the metabolomics analysis in fungi research. I agree with the overall structure and contents of the manuscript.
However, it requires huge improvement for the English writing. For examples: many long sentences could be divided to short sentences to make them easier to be understood. A lot of gramma mistakes need to be corrected. All the species names should be wrote as italic fonts. Legend of Fig. 2 needs to be modified. Only 4 'aspects' have been listed in Line 12.
Author Response
Thank you for your valuable and thoughtful comments. We have made correction according to the Reviewer’s comments.
- Responds: We have carefully checked and improved the English writing in the revised manuscript.
- Responds: All species names have been revised to italicized font.
- Responds: Thank you very much for your advice, the legend for Figure 2 has been modified ( page 11, lines 564-566).
- Responds: We are very sorry for our negligence on the abstract content, we have already made it up for the content (page 1, line 12).
Reviewer 3 Report (New Reviewer)
The paper submitted by Li and coworkers is a review on the application of metabolomics in the research focused on different fungi. The manuscript is well prepared and is very interesting. In my opinion it should be published in Molecules after addressing some minor comments:
1) please provide figure showing chemical structures of fungal metabolites discovered by the metabolomics approach (compounds mentioned in section 3.3) It would be good is each compounds is accompanied by its molecular formula and weight for the readers convince and info in which fungus it was detected
Author Response
Special thanks to you for your good comments. We strongly agree with the reviewers' suggestions and have added a table to the text (page 27-29, lines 1072-1075).
Round 2
Reviewer 2 Report (New Reviewer)
I saw huge improvement of the English writing. However, I suggest the editor to give authors extra time to further improve the possible grammatical and format issues. Below are some examples:
Line 407 can be shorted as "Agaricus bisporus is a worldwide edible mushroom"
Line 428, missing literature reference.
Line 611-613, the description is not very accurate. There are some databases available for fungal secondary metabolites, and software for data analysis piplines, although they are not perfect.
Author Response
Comments: I saw huge improvement of the English writing. However, I suggest the editor to give authors extra time to further improve the possible grammatical and format issues.
Response: Thank the reviewers for their kind comments. We invited a native English speaking colleague to review and revise our manuscript, and carefully corrected the errors in the article.
Comments: Line 407 can be shorted as "Agaricus bisporus is a worldwide edible mushroom"
Response: Thanks to the reviewers for the example, we have made further revisions to the language of the full article. For modifications herein see (page 9, line 639)
Comments: Line 428, missing literature reference.
Response: We are sorry that we wrote the wrong author of the content here, which belongs to the same artical as the previous sentence. We have revised the content here, see (page 9, lines 654-658)
Comments: Line 611-613, the description is not very accurate. There are some databases available for fungal secondary metabolites, and software for data analysis piplines, although they are not perfect.
Response: We strongly agree with the reviewers and have made changes to the non rigorous statements (page 14, lines 1011-1013)
This manuscript is a resubmission of an earlier submission. The following is a list of the peer review reports and author responses from that submission.
Round 1
Reviewer 1 Report
I read with interest this manuscript, and regret to inform the authors and editor that do not support the publication of the manuscript in its present form.
The addressed subject has been already revised by others ( see examples here https://pubmed.ncbi.nlm.nih.gov/?term=metabolomics+fungal&filter=pubt.review&sort=date; https://scholar.google.com/scholar?hl=es&as_sdt=0%2C5&q=metabolomics+fungal&btnG=), and currently, the manuscript does not add valuable information to the field. The proposed manuscript structure is interesting, but all the sections provide information in a shallow way and. as a consequence, it is not an authoritative text when compared with other recent review papers. My suggestion is to focus on one particular aspect, perhaps the fungal response to stress is the one less addressed in the last five years, and to provide a thorough literature revision.
Finally, the conclusion section is just a summary of the main text and as it is, could be deleted without affecting the manuscript content. The idea of the conclusion/concluding remarks section is to provide the authors' opinion about the significant progress achieved in the last years in a particular area, and most importantly, to provide an opinion about the relevant challenges to overcome and the way the scientific community should address research on this field in the coming years.
Reviewer 2 Report
The review article submitted by Jian et al., selected a timely topic. But for my understanding, the article needs more work. There are a high number of content editing typos detected during my review. I have highlighted a few of them. I encourage the authors to go through again and correct it.
I have few specific comments for the authors,
1. the authors have divided the topic into a few subtopics and each section poorly discussed. Need more effort on those sections since this is a review. Especially, sample preparation, analytical methods - make a summary of all the possible previous work and how a sample should prepare for an efficient and effective metabolomics study, what analytical methods used, what configurations are described so far etc.
2. Is there any pre-treatment needs before the analysis, how and what?
3. Is derivatization needed for GCMS? there are metabolomic studies done without derivatizing the sample? check lit.
4. analytical methods - GC, GCMS, LC, LCMS, LCHRMS, QTOF, MALDI, HR-magic horn rotation, ion monitoring are few analytical techniques authors mentioned in the review. Its better to give a basic idea of those methods. In the form of a table with conditions - ex- LCMS conditions (LC solvent systems, columns, MS- polarity, energy, fragments, ionization, ESI, APCI, etc..)
5. discuss the terms- metabolomics and metabonomics?
6. Line 218 - rephrase
7. 220 - 4th benzoic acid derivative? folic acid?
8. Natural product section is poorly written - Authors discuss the ry metabolites in the initial part but natural products are 2ry metabolites. I don't see any connectivity of this section 3.3. Encourage authors to re-write.
9. 3.4 section should follow the same format of the paper.
10. There are new techniques in metabolomics such as LC-NMR, qNMR, authors should include new techniques to the review.
11. Authors should try to express a message in each section. Citing others' work is not enough. Mention what Authors think about each section and what suggestions could give or propose.
12. reference list double numbers.
Since the manuscript needs more work and effort, in this current stage I am unable to accept this manuscript for the publication.
thanks
